# Investigating Intensity and Percentage of p53 Nuclear Expression in Prostate Cancer: Findings from a Cohort of U.S. Military Veterans

**DOI:** 10.3390/cancers17061004

**Published:** 2025-03-17

**Authors:** William R. Gesztes, Coen J. Lap, Rithika Rajendran, Maryam M. Dalivand, Guoqing Diao, Shanshan Liu, Maneesh Jain, Victor E. Nava

**Affiliations:** 1Department of Pathology, Washington DC VA Medical Center, Washington, DC 20422, USA; dr.rithikarajendran@gmail.com (R.R.); victor.nava@va.gov (V.E.N.); 2Department of Pathology, The George Washington University Hospital, Washington, DC 20037, USA; 3Department of Pathology and Laboratory Medicine, Memorial Sloan Kettering Cancer Center, 1275 York Avenue, New York, NY 10065, USA; 4The Edward P. Evans Precision Oncology Center of Excellence, Washington DC VA Medical Center, Washington, DC 20422, USAmaneesh.jain@va.gov (M.J.); 5Department of Hematology and Oncology, Lombardi Comprehensive Cancer Center, Georgetown University, Washington, DC 20007, USA; 6Department of Biostatistics and Bioinformatics, Milken Institute School of Public Health, The George Washington University, Washington, DC 20037, USA; gdiao@gwu.edu (G.D.); shanshanliu@gwu.edu (S.L.)

**Keywords:** prostate cancer, p53, *TP53*, biomarker, immunohistochemistry, African American, progression

## Abstract

The role of p53 immunohistochemistry (IHC) in predicting disease progression in prostate cancer (PCa) was investigated, revealing an unmet need to better understand the interpretation of p53 nuclear expression. This study was conducted at the DC VA Medical Center and involved 84 patients (majority self-identified as African American) with PCa diagnosed between 1996 and 2021. Nuclear expression of p53 was scored in various specimens according to the intensity (0, 1+, 2+, 3+) and percentage (0%, <1%, 1–5%, >5%) of p53-positive tumor cells. In total, 40% of patients were found to have p53 expression, with 21% showing maximum intensity (3+). Maximum intensity was significantly associated with a higher Grade Group (*p* < 0.001), elevated PSA levels at the time of biopsy (*p* < 0.001), biochemical recurrence (*p* < 0.001), and metastasis (*p* < 0.001). Notably, only patients who developed metastatic disease were found to have maximum p53 intensity. This study underscores the importance of evaluating p53 nuclear intensity, in addition to percentage p53 positivity, to predict disease progression in PCa.

## 1. Introduction

Prostate cancer (PCa) remains the most commonly diagnosed non-cutaneous cancer among men in the United States and the second leading cause of cancer-related death [1]. However, the clinical outcome is variable, from indolent to aggressive disease that is metastatic at time of presentation [2,3]. Although many patients experience a relatively protracted clinical course, a distinct subset of patients, even when diagnosed early, succumb to their disease prematurely. Historically, the challenge for clinicians has been to better risk-stratify patients in hopes of avoiding either under- or overtreatment both at the time of diagnosis and/or after follow-up interventions [4]. Despite extensive research to identify biomarkers in PCa, few studies have garnered significant clinical traction to address this unmet, yet critical, need [5,6].

The p53 protein mediates tumor suppression functions by regulating DNA repair, cell cycle arrest, apoptosis, autophagy, homeostasis, and cell senescence [7]. Accumulating nuclear p53 as a result of DNA damage activates genes that initiate cell cycle arrest and p53-mediated apoptosis. As such, functioning p53 is crucial for maintaining genomic integrity and preventing malignant transformation. *TP53* is found to be altered in over 50% of human cancers, which has been causatively linked to both cancer initiation and progression, including the promotion of metastatic disease [8]. Increased utilization of next-generation sequencing (NGS) has revealed the presence of recurrent alterations in *TP53* in approximately 6–7% of patients with localized PCa and up to 50% of patients with advanced disease [9,10,11]. However, observed rates of *TP53* alterations in PCa vary, as indicated by a report of racial differences in *TP53*, where mutations occurred less frequently in African American males versus those of European descent [12]. Although the exact impact of specific *TP53* alterations in PCa remains to be determined, loss of tumor-suppressive functions of wild-type p53 and mutant p53 gain of function have been described, including direct cross-talk between p53 and androgen receptor signaling pathways [13,14]. Prior studies have shown an association between *TP53* alterations in primary prostate tumors and disease recurrence and progression [15,16,17,18,19,20]. Moreover, the presence of mutations in *TP53* in PCa is associated with shorter overall survival [21].

Therefore, determining a patient’s *TP53* mutational status is prognostically relevant in PCa. However, sequencing every tissue specimen is neither desirable nor an optimal use of resources. p53 immunohistochemistry (IHC) has been shown to be a good surrogate marker for underlying mutations in *TP53* [22,23,24,25]. We have shown before that in radical prostatectomy (RP) specimens from patients with PCa, high levels of p53 nuclear expression was suggestive of the presence of pathogenic *TP53* alterations, and, together with lymphovascular invasion (LVI) status, enhances the early prognostication of PCa progression and development of metastatic disease [26]. Thus, p53 IHC staining is a conceivable, cost-effective screening tool for molecular testing, as in other malignancies [27]. While guidelines for interpreting aberrant p53 expression exist in other tumors, a broad consensus is currently lacking in PCa, necessitating further investigation [28,29]. Although some studies have used the percentage of p53 nuclear expression as a threshold for overexpression, others have prioritized nuclear intensity or a combination of both [16,22,26,30,31,32]. Hence, before such a biomarker can be routinely implemented in PCa, a tumor-specific standard for p53 overexpression is necessary. Therefore, this study aims to compare two fundamental approaches for assessing p53 nuclear expression across a range of specimen sources with regard to PCa progression, in an effort to achieve a more optimal interpretation of p53 IHC findings.

## 2. Methods and Materials

### 2.1. Patient Selection

We randomly selected 84 patients diagnosed with PCa between 1996 and 2021 at the Washington DC Veterans Affairs Medical Center (DC VAMC). The resulting cohort reflects a representative cross-section of patients receiving care at this single VA facility. Patients were included if adequate formalin-fixed paraffin-embedded (FFPE) tumor tissue was available. Clinical information, including age, race/ethnicity, Gleason score, Grade Group, PSA at time of biopsy, development of biochemical recurrence (BCR), disease stage at diagnosis, and development of metastatic disease, was retrieved. Localized low risk was defined as AJCC stage I, locally advanced disease as AJCC stage II and III, and metastatic disease as AJCC stage IV. This study was approved by the DC VAMC Institutional Review Board IRBNet#1651999 on 28 October 2021.

### 2.2. Tumor Tissue Selection

Representative sections of random (non-targeted) prostate core biopsies, RPs, transurethral resections of prostate (TURPs), and distant metastatic deposits were included. A representative formalin fixed paraffin-embedded (FFPE) block from each patient was selected. Index (dominant) tumors from aforementioned specimens were selected upon histological examination of hematoxylin and eosin (H&E) slides.

### 2.3. Immunohistochemistry

Consecutive four-micron-thick unstained tumor sections were prepared for immunostaining with p53 (anti-p53 mouse monoclonal antibody DO-7, Biocare Medical, Pacheco, CA, USA), and anti-podoplanin antibody (D2-40, Biocare Medical) was used to identify LVI. Selected slides were also stained with CD34 (QBEND 10, Agilent DAKO, Santa Clara, CA, USA) and CD31 (JC70A, Agilent DAKO, Santa Clara, CA, USA) to determine LVI. p53 nuclear expression was manually scored according to the highest intensity observed (0 absent, 1+ light, 2+ medium, or 3+ maximum) and the percentage of tumor cells expressing any level of intensity of the aggregate index tumor area (0%, <1%, 1–5%, and >5%) at 200× magnification (Figure 1).

A p53 range of >5% or less was applied based on results from a prior study [26]. Rare cases with exclusive cytoplasmic staining were considered negative, equivalent to “absent” or “0%” p53 expression. LVI was considered positive when tumor cells were identified within spaces lined by the endothelium, delineated by anti-podoplanin (D2-40), CD31 or CD34 immunoreactivity. All slides were reviewed independently by two pathologists (W.R.G, V.E.N) who were blinded to the clinical data. Discrepancies between the two pathologists were resolved by consensus.

### 2.4. Mutational Analysis

DNA NGS was performed on FFPE tissue at Foundation Medicine or at our institution to identify *TP53* alterations in a subset of patients with metastatic disease. Results were available for a select number of patients (*n* = 29).

### 2.5. Statistical Analysis

Descriptive statistics are reported for demographics and clinical characteristics. For continuous variables, mean and standard deviation (SD) are reported if they are normally distributed. Otherwise, mean, and interquartile range (IQR) are reported. For categorical variables, frequency and percentage are reported. Chi-square tests or Fisher’s exact tests (when the expected count was <5) were used to evaluate the association between p53 intensity and categorical variables, including race/ethnicity, D2-40 LVI positivity, Gleason score, Total Gleason Score, Grade Group, BCR, development of metastatic disease, and presence of *TP53* mutations. Two-sample *t*-tests were used to evaluate the association between p53 intensity and variables including age and PSA at diagnosis. The statistically significant level of all the tests was set to 0.05. All statistical analyses were performed using R Statistical Software (version 4.2.2).

## 3. Results

Relevant clinical data, such as the following, were available for analysis from a total of 84 patients, including index tumors: 50 core biopsies, 14 RPs, 7 TURPs, and 13 metastatic deposits. Demographic information and clinical characteristics are shown in Table 1.

The mean age at diagnosis for the complete cohort was 64.1 years (SD 7.48 years), and 63 out of the 84 patients (75%) self-identified as African American. In total, 16 patients (19.1%) were found to have Grade Group 1 disease, 19 patients (22.6%) Grade Group 2, 4 patients (4.8%) Grade Group 3, 20 patients (23.8%) Grade Group 4, and 12 patients (14.3%) Grade Group 5. For 13 patients (15.5%), the Grade Group was not applicable due to the convention of not grading metastatic foci. The median PSA at the time of biopsy was 11.6 ng/mL (IQR 6.35–73.60). The median time of follow-up was 6.2 years for all patients.

The dynamic range of p53 immunohistochemical signals observed in tumor nuclei at various percentages of expression is shown in Table 2.

In general, the intensity of the signal was associated with the percentage of expression and with the Grade Group of the tumor. All patients with >5% p53 expression of the tumor showed at least one malignant nucleus with maximum (3+) intensity. Overall, 34 tumors (40% of all tumors) exhibited p53 nuclear expression, of which 18 (21.4% of all tumors) showed maximum (3+) intensity regardless of the percentage of positive nuclei. Of these 18 tumors, 4 were obtained from patients with locally advanced (AJCC stage II or III) PCa at the time of diagnosis, and 13 were obtained from those with de novo metastatic disease after further work-up (Table 1). When the percentage of positive nuclei was considered as well, 7 out of the 18 tumors (39%) with maximum (3+) intensity were found to have expression detected in at least 5% of tumor cells. Notably, 11 of 18 (61%) patients with maximum (3+) intensity displayed immunoreactivity in less than 5% of tumor cells. The presence of maximum (3+) intensity p53 expression was not associated with the type of specimen (core biopsy, RP, TURP, or distant metastatic deposit) (Table 1; *p* = 0.468).

No statistical differences were observed between the presence or absence of maximum (3+) p53 staining and age at diagnosis or ethnicity (Table 1). Importantly, a significant difference was observed between Grade Group and p53 intensity (*p* < 0.001). Maximum (3+) expression for p53 was associated with a higher Grade Group, and all cases with maximum (3+) intensity were Grade Group 3 or higher. In contrast, patients with low expression of p53 (1+ or 2+ intensity) were more frequently in a lower Grade Group (1 or 2). In addition, maximum (3+) intensity for p53 was associated with a higher PSA level at the time of biopsy (*p* < 0.001). While patients with a maximum (3+) p53 intensity showed a median PSA level of 172 ng/mL (IQR 26.19–700), the value was 8.30 ng/mL (IQR 6.10–20.20) for patients with less-than-maximum p53 intensity. In line with these results, the occurrence of BCR in patients without metastatic disease at diagnosis was more frequent with a maximum (3+) intensity of p53 (27.8% vs. 22.7%, *p* < 0.001). When using maximum (3+) intensity as the sole criterion for p53 overexpression, a significant difference was observed between metastatic and non-metastatic patients. Specifically, 18 out of 46 (39%) tumors from patients with metastasis on biopsy, or those who developed metastatic disease, were positive for maximum (3+) p53 staining, compared to 0 out of 38 (0%) tumors from patients without metastatic disease (*p* < 0.001) (Table 2). Interestingly, when maximum (3+) p53 intensity was combined with expression in greater than 5% of tumor cells to determine overexpression, the difference between the two groups remained significant (*p* = 0.015). However, the percentage of p53 overexpressed metastatic tumors was reduced to 15% (7 out of 46 tumors). Importantly, maximum (3+) p53 staining was identified only in patients who were diagnosed with, or eventually developed, metastatic disease.

*TP53* sequencing data were available for 29 patients, 27 with metastatic disease and 2 without. A total of 12 mutations in *TP53* were identified in 10 patients. Corresponding *TP53* sequencing data with IHC results are shown in Table 3.

All alterations were observed in tumor specimens from patients who were diagnosed with, or eventually developed, metastatic disease. The majority of mutations (10/12) were missense variants located in the DNA-binding domain of *TP53* (Figure 2).

The other two mutations observed in two patients were frameshift deletions in the N-terminal domain resulting in truncated p53 (null mutations) that failed to be recognized by the antibody we used. Interestingly, almost all patients with an in-frame missense *TP53* variant showed maximum (3+) intensity by IHC, and the majority expressed p53 in over 5% of tumor cells. Indeed, only one patient with the missense R273H showed an absence of p53 expression (score 0), which could be explained by the extremely low variant allelic frequency (0.13%) observed. Statistical analysis revealed a significant association between the presence of a missense *TP53* variant and maximum (3+) IHC intensity (*p* < 0.01) in the mutated subset (Table 4). This significant association persisted when the level of p53 expression was present in greater than 5% of tumor cells (*p* < 0.01). Due to the limited number of sequenced tumors, we cannot definitively conclude whether maximum (3+) IHC intensity can be a surrogate for the presence of *TP53* mutations. However, our data demonstrate that the presence of missense mutations is significantly associated with maximum (3+) p53 expression on IHC.

In summary, the presence of maximum (3+) p53 nuclear expression, regardless of percentage, was found to be significantly associated with a higher Grade Group (*p* < 0.001), higher PSA at the time of biopsy (*p* < 0.001), the development of BCR (*p* < 0.001) and metastatic disease (*p* < 0.001). Moreover, the presence of missense mutations in *TP53* was found to be significantly associated with the presence of maximum (3+) p53 intensity, even when the percentage of tumor cells was less than 5%.

## 4. Discussion

The tumor suppressor *TP53* is one of the most frequently mutated genes in malignant tumors and has been found to be crucial for tumor initiation, progression and metastasis [34]. Although alterations in *TP53* were initially thought to be late events in PCa oncogenesis and associated with androgen resistance, deep molecular profiling of metastatic castration-sensitive PCa and even localized PCa highlighted the existence of recurrent genomic alterations in *TP53* as well [31,35,36]. Conveniently, IHC can be used as an acceptable surrogate to identify *TP53* alterations. However, a lack of standardization along with variability in prior study designs has hampered the implementation of p53 IHC as a prognostic marker and screening tool in PCa.

In the present study, two fundamentally different, and commonly applied, approaches to assess p53 nuclear expression were evaluated as predictors of PCa progression, namely intensity (0, 1+, 2+, 3+) observed and percentage (0%, <1%, 1–5%, and >5%) of tumor cells expressing any level of intensity. We observed that the presence of maximum (3+) p53 nuclear intensity by IHC was significantly correlated with predictors of worse outcomes (Grade Group, Gleason score, higher PSA level at diagnosis, and development of BCR and metastasis) regardless of the percentage of p53 nuclear expression. While prior studies did not always share the same methodology for evaluating p53 expression in PCa, many, if not most, consistently showed that “positive”, “overexpressed”, “altered”, or “aberrant” p53 expression is associated with more aggressive disease, which is consistent with our findings [16,26,37,38]. Studies that lack this association tend to use overly restrictive criteria for labeling p53 expression as being overexpressed [39,40]. Due to the prostate being a less cellularly active organ compared to ovaries, for example, it is prudent to conclude that the percentage threshold for p53 immunoreactivity is inherently lower as well, as p53 activity tends to be directly correlated with proliferation rate [41,42].

The intensity of p53 on IHC proved to be strongly associated with disease progression, more so than percent tumor expression. This result can at least be partially explained by sampling, as the majority of our specimens represented random core biopsies, which may underestimate overall p53 expression. RP specimens are likely superior for estimating the true percentage of aberrant p53 expression, as they are more representative of the lesion and minimize sampling bias [43]. However, core biopsies usually provide the primary diagnosis, and using p53 intensity may be useful to predict worse outcome independently of percent tumoral expression, as suggested previously [32]. Our observation that maximum nuclear intensity was detected only in patients with metastatic disease at diagnosis, or patients who would develop metastatic disease during follow-up, highlights the potential of this marker to inform us about the severity and likelihood of the progression of disease independently of other pathology-related factors. Although the number of patients in this study is too small to arrive at a definitive conclusion, these findings suggest that detecting maximum (3+) p53 nuclear intensity should raise concerns that the patient has, or will develop, advanced disease during follow-up.

Furthermore, recognition of maximum (3+) p53 nuclear intensity by IHC, regardless of percent tumoral positivity, appears to be a practical approach that may benefit the routine practice of surgical pathology. In our study, concordant p53 expression levels for maximum nuclear intensity reached 94% of cases between our two pathologists, who have varying degrees of experience (>25 years and <5 years). Discrepancies were resolved with follow-up consensus in 100% of cases. In contrast, exact determination of percent expression over a range of intensities would likely place unacceptable demands on routine clinical practice unless automated imaging analysis is introduced. However, when a single numerical percentage cut-off of >5% was applied, concordance for this criterion increased to greater than 90%. In the future, automated computer analysis of digitalized pathology images may circumvent the current limitations of inter-observer variability and labor-intensive estimation of tumor expression to better evaluate tumoral p53 expression in PCa, which is known to be focal and variable.

We showed that conducting a manual estimation of p53 nuclear expression using the primary criterion of maximum (3+) p53 nuclear intensity is a feasible and practical approach. Owing to the focal nature of p53 expression, we conclude that in day-to-day practice, optimal screening of PCa with p53 IHC should not be limited to the index tumor alone, and should ideally include examination of additional higher grade tumor foci. Of note is that the absence of p53 nuclear expression in tissue specimens other than RP should not by itself automatically raise concerns of a possible *TP53* deletion (null mutation) owing to the aforementioned focal expression of p53. However, a complete absence of p53 expression in RP specimens, after careful review in cases with higher grade (GG4, GG5) lesions, may raise the possibility of a null *TP53* mutation (frameshift, truncating mutations). Moreover, prior research has shown that prostate tumors can be completely negative for p53 IHC staining, yet still harbor underlying *TP53* mutations [37]. Although the incidence of these “silent” mutations is small and their impact is unclear, they do highlight the limitations of p53 IHC. A proposed interpretation of p53 nuclear expression in PCa, shown in Table 5, considers the aforementioned observations from this study.

As alluded to earlier, our study is limited by its skewed representation of tissue sources examined (i.e., majority core biopsies) and its moderate sample size, which possibly also explains the lack of statistical significance of LVI in our cohort. The fact that p53 IHC nuclear expression, even with a moderate sample size, was highly associated with disease progression speaks to its overall weight and importance. Other limitations include the retrospective nature of this study and pathology review performed by two pathologists with a close working history. Another important weakness worth discussing is the integrity of RP specimens. As is common practice, RP specimens are quartered to fit an entire cross-section on standardized glass slides. This practical consideration complicates the evaluation of the overall percentage of p53 expression when a tumor spans multiple glass slides, resulting in possible over- or underestimation. Careful and precise percentage estimation is more accurate when performed on whole-mount slides. However, the use of whole-mount processed RP specimens, in general, is limited. In this sense, our review of quartered RP specimens can be viewed as a strength as it more accurately reflects the reality most pathologists face in everyday practice. In terms of sequencing data, molecular testing is heavily skewed towards patients who have developed metastatic disease and lacks the ability for comprehensive testing of negative or low-expressing p53 tumors. A final weakness of our study involves its lumping of cytoplasmic p53 expression together with the negative category. While extremely rare in PCa, the biologic implications of cytoplasmic expression is not well understood [44,45]. Topics not specifically addressed are the issues of null p53 expression and how to select cases deserving of p53 IHC staining. With regard to the latter, further studies are needed to identify patients who may benefit the most from incorporating this biomarker. Hypothetically, patients classified as having Grade Group 3 disease and/or belonging to the intermediate risk category could qualify for p53 IHC to improve the stratification of these historically challenging groups of patients.

The fact that our cohort is predominantly composed of patients who identified as African American could be seen as both a strength and a limitation. Most prior studies reviewing p53 IHC in PCa have had inversed proportions of African American and Caucasian patients, raising questions about the prevalence of p53 IHC expression in this minority population [12,46,47]. One study did observe that *TP53* is less frequently altered in African American patients when compared to patients of European ancestry [33,48]. Interestingly, we found no difference in p53 expression (intensity or percentage) between African American patients and other ethnicities included in our cohort, suggesting that p53 may be a valid biomarker for this minority population.

## 5. Conclusions

Maximum (3+) p53 nuclear intensity is a valuable prognostic marker of PCa progression, regardless of percentage positivity and independent of specimen source. Therefore, p53 overexpression, as characterized in this study, may serve as a useful screening tool for improved risk stratification and optimized patient care in a subset of patients.

## Figures and Tables

**Figure 1 cancers-17-01004-f001:**
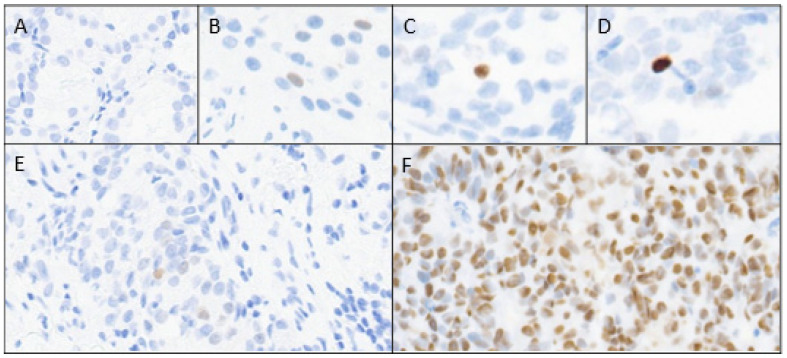
p53 nuclear expression scoring; intensity (**A**–**D**) and percentage (**E**,**F**) of tumor. (**A**) 0 (absent); (**B**) 1+ (light); (**C**) 2+ (medium); (**D**) 3+ (maximum); (**E**) ≤5%; (**F**) >5%.

**Figure 2 cancers-17-01004-f002:**
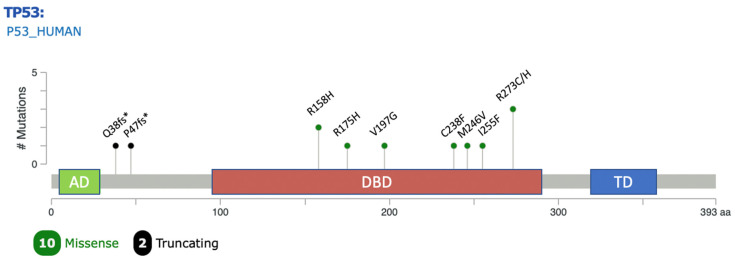
Lollipop plot showing position and frequency of mutations in *TP53* observed in the study across protein domains. Plot generated using the Mutation Mapper tool in cBioportal [33]. AD: transactivation domain; DBD: DNA binding domain; TD: tetramerization domain.

**Table 1 cancers-17-01004-t001:** Patients’ demographic and clinical characteristics, stratified by p53 staining intensity.

	Overall *n* = 84	p53 Absent (0), Light (1+) or Medium (2+) Intensity *n* = 66	p53 Maximum Intensity (3+) *n* = 18	*p* Value
**Mean age at diagnosis (STD)**	64.1 (7.48)	63.3 (7.01)	67.1 (8.59)	0.0561
**Race/Ethnicity**				1.0000
African American	63 (75%)	49 (74.2%)	14 (77.8%)	
Caucasian	20 (23.8%)	16 (24.2%)	4 (22.2%)	
Native Hawaiian	1 (1.2%)	1 (1.5%)	0 (0%)	
**D2-40 LVI (*n* = 42)**				0.0755
Positive	8 (19.1%)	4 (12.5%)	4 (40%)	
Negative	34 (80.9%)	28 (87.5%)	6 (60%)	
**Gleason Score**				<0.001
3 + 3 (6)	16 (19.1%)	16 (24.2%)	0 (0%)	
3 + 4 (7)	19 (22.6%)	19 (28.8%)	0 (0%)	
4 + 3 (7)	4 (4.8%)	3 (4.6%)	1 (5.6%)	
3 + 5 (8)	1 (1.2%)	1 (1.5%)	0 (0%)	
4 + 4 (8)	19 (22.6%)	14 (21.2%)	6 (27.8%)	
4 + 5 (9)	11 (13.1%)	7 (10.6%)	4 (22.2%)	
5 + 4 (9)	1 (1.2%)	0 (0%)	1 (5.6%)	
n/a (Metastatic deposits)	13 (15.5%)	6 (9.1%)	7 (38.9%)	
**Total Gleason Score**				<0.001
6	16 (19.1%)	16 (24.2%)	0 (0%)	
7	23 (27.4%)	22 (33.3%)	1 (5.6%)	
8	20 (23.8%)	15 (22.7%)	5 (27.8%)	
9	12 (14.3%)	7 (10.6%)	5 (27.8%)	
n/a (Metastatic deposits)	13 (15.5%)	6 (9.1%)	7 (38.9%)	
**Grade Group**				<0.001
Grade Group 1	16 (19.1%)	16 (24.2%)	0 (0%)	
Grade Group 2	19 (22.6%)	19 (28.8%)	0 (0%)	
Grade Group 3	4 (4.8%)	3 (4.6%)	1 (5.6%)	
Grade Group 4	20 (23.8%)	15 (22.7%)	5 (27.8%)	
Grade Group 5	12 (14.3%)	7 (10.6%)	5 (27.8%)	
n/a (Metastatic deposits)	13 (15.5%)	6 (9.1%)	7 (38.9%)	
**Median PSA at diagnosis (IQR)**	11.60 (6.35–73.60)	8.30 (6.10–20.20)	172 (26.19–700)	<0.001
**BCR**				<0.001
No	34 (40.5%)	33 (50%)	1 (5.6%)	
Yes	20 (23.8%)	15 (22.7%)	5 (27.8%)	
Unknown	1 (1.2%)	1 (1.5%)	0 (0%)	
n/a	29 (34.5%)	17 (25.8%)	12 (66.7%)	
**Disease stage at time of diagnosis**				<0.001
Localized low risk (AJCC stage I)	14 (16.7%)	14 (21.2%)	0 (0%)	
Locally advanced cohort (AJCC stage II and III)	24 (28.6%)	20 (30.3%)	4 (22.2%)	
Metastatic cohort (AJCC stage IV)	46 (54.8%)	32 (48.5%)	14 (77.8%)	
**Type of specimen**				
Core Biopsy	50 (59.6%)	45 (68.2%)	5 (27.8%)	
Radical Prostatectomy	14 (16.7%)	11 (16.7%)	3 (16.7%)	
TURP	7 (8.3%)	4 (6.1%)	3 (16.7%)	
Distant Metastatic Deposits	13 (15.5%)	6 (9.1%)	7 (38.9%)	

BCR, biochemical recurrence; STD, standard deviation; LVI, lymphovascular invasion; IQR, interquartile range; AJCC, American Joint Committee of Cancer; TURP (transurethral resection of prostate).

**Table 2 cancers-17-01004-t002:** p53 nuclear IHC expression profile in metastatic and non-metastatic disease categories.

	Overall *n* = 84	Non-Metastatic Disease *n* = 38	Metastatic Disease *n* = 46	*p* Value
**p53 nuclear intensity alone**				<0.0001
3+ (maximum) intensity	18 (21.4%)	0 (0%)	18 (39.1%)	
0 (absent), 1+ (light) or 2+ (medium) intensity	66 (78.6%)	38 (100%)	28 (60.9%)	
**p53 nuclear combined score**				0.0146
3+ (maximum) intensity and >5%	7 (8.3%)	0 (0%)	7 (15.2%)	
0 (absent), 1+ (light) or 2+ (medium) intensity and ≤5%	77 (91.7%)	38 (100%)	39 (84.8%)	

IHC, immunohistochemistry.

**Table 3 cancers-17-01004-t003:** *TP53* mutation and p53 IHC pattern in patients with prostate cancer.

No.	Disease Stage	Grade Group	Effect	Genomic Position	Nucleic Acid Alteration	Amino Acid Alteration	Location Within *TP53*	Type	Clinical Significance (ClinVar)	VAF (%)	IHC
1	Metastatic	4	Missense variant	17:7675088	*c.524G* > *A*	p.R175H	DNA binding domain	SNV	Pathogenic or likely pathogenic	13.2	+3, <1%
2	Metastatic	4	Missense variant	17:7674941	*c.590T* > *G*	p.V197G	DNA binding domain	SNV	Uncertain significance	83.7	+3, >5%
3	Metastatic	5	Frameshift variant	17:7676257	*c.112del*	p.Q38fs * 6	N-terminal domain (TAD)	Del	Pathogenic or likely pathogenic	22.3	0
4	Metastatic	2	Missense variant	17:7673802	*c.818G* > *A*	p.R273H	DNA binding domain	SNV	Pathogenic or likely pathogenic	0.13	0
5	Metastatic	n/a ^I^	Frameshift variant	17:7676229	*c.140del*	p.P47fs * 76	N-terminal domain(TAD)	Del	Pathogenic or likely pathogenic	34.1	0
6	Metastatic	n/a ^I^	Missense variant	17:7674250	*c.713G* > *T*	p.C238F	DNA binding domain	SNV	Pathogenic or likely pathogenic	64.7	+3, >5%
7	Metastatic	5	Missense Variant Missense variant	17:7673803 17:7675139	*c.817C* > *T* *c.473G* > *A*	p.R273C p.R158H	DNA binding domain DNA binding domain	SNV SNV	Pathogenic or likely pathogenic (both)	37 30	+3, >5%
8	Metastatic	4	Missense variant	17:7674200	*c.763A* > *T*	p.I255F	DNA binding domain	SNV	Conflicting reports	40.3	+3, <1%
9	Metastatic	4	Missense Variant Missense variant	17:7673803 17:7675139	*c.817C > T* c.473G > A	p.R273C p.R158H	DNA binding domain DNA binding domain	SNV SNV	Pathogenic or likely pathogenic (both)	32 28	+3, >5%
10	Metastatic	n/a ^I^	Missense variant	17:7674227	*c.736A > G*	p.M246V	DNA binding domain	SNV	Pathogenic or likely pathogenic	44.8	+3, >5%

VAF, variant allele frequency; IHC, immunohistochemistry; SNV, single nucleotide variant; Del, deletion. I: n/a = denote metastatic deposits which are not graded by convention. *: the correct scientific coding of the specific amino acid alteration.

**Table 4 cancers-17-01004-t004:** Association between *TP53* mutational status and p53 nuclear IHC expression profile. For the analysis, the two patients with a frameshift deletion were excluded as this resulted in a null mutation.

	Overall *n* = 29	*TP53* Mutation Detected	No *TP53* Mutation Detected	*p* Value
**p53 nuclear intensity alone**				<0.01
3+ (maximum) intensity	13 (44.8%)	7	6	
		(87.5%)	(28.6%)	
0 (absent), 1+ (light) or 2+ (medium) intensity	16 (55.2%)	1	15	
		(12.5%)	(71.4%)	
**p53 nuclear combined score**				<0.01
3+ (maximum) intensity AND >5%	6	5	1	
	(20.7%)	(62.5%)	(4.8%)	
0 (absent), 1+ (light) or 2+ (medium) intensity AND ≤5%	23	3	20	
	(79.3%)	(37.5%)	(95.2%)	

IHC, immunohistochemistry.

**Table 5 cancers-17-01004-t005:** Proposed interpretation of p53 IHC expression in prostate cancer. Null p53 expression not accounted for in either RP (**A**) or non-RP (**B**) specimens. (**A**) Interpretation of IHC p53 expression profile in radical prostatectomy specimens. (**B**) Interpretation of IHC p53 expression profile in non-radical prostatectomy specimens.

(**A**)	
Characterizing p53 nuclear expression status in RP specimens	
Low expression	1+ (light) or 2+ (medium) nuclear intensity AND ≤5%
Equivocal expression	1+ (light) or 2+ (medium) nuclear intensity AND >5%
Overexpression	3+ (maximum) nuclear intensity of any percent
(**B**)	
Characterizing p53 nuclear expression status in non-RP specimens *	
Low expression	1+ (light) nuclear intensity
Equivocal expression	2+ (medium) nuclear intensity
Overexpression	3+ (maximum) nuclear intensity

* Includes prostate core biopsies, TURPs, metastatic deposits; IHC, immunohistochemistry; RP, radical prostatectomy.

## Data Availability

Data are contained within the article.

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
