# Peer review of "Investigating Intensity and Percentage of p53 Nuclear Expression in Prostate Cancer: Findings from a Cohort of U.S. Military Veterans"

_cancers, 2025, doi:10.3390/cancers17061004_

Round 1

Reviewer 1 Report (Previous Reviewer 1)

Comments and Suggestions for Authors

All previous comments have been addressed, and the manuscript is now fully revised.  It is recommended for consideration for publication in its present form

Author Response

We wish to sincerely thank Reviewer #1 for their earlier constructive suggestions and final approval of our re-submitted manuscript for publication after the first round of peer review.   

Reviewer 2 Report (Previous Reviewer 2)

Comments and Suggestions for Authors

The authors presented that TP53 immunohistochemical staining intensity is important for detecting poor risk prostate cancer disease focusing cohort mostly composed by African American.  However, there are several flaws that should be re-considered.

  • The title “predominantly Black patient cohort” is not scientific and confusing. The authors should focus on African American cohort only and revise the title.
  • The authors presented that they could not determine Grade Group of metastatic lesions. Why? Were the samples derived from post-hormonal treatment?
  • Table 1 should include information about samples derived from. How many cases were localized cases or metastatic cases?  The authors should also include information about the number of tissues derived from: RP, TUR, and biopsy.
  • BCR depends on how long the patients are followed. Fisher’s exact test is not appropriate for the analysis.  The authors should use time-dependent analysis such as logrank test using KM survival curve.
  • The authors should present clinical and pathological information about cases included in Table 3.
  • “Our observation that maximum nuclear intensity was detected only in patients who developed metastatic disease, regardless of type of specimen (core biopsy, RP, TURP, or distant metastatic deposit), highlights the potential of this marker to inform about the severity and likelihood of progression of disease independent of other pathology-related factors” This information is important but we should know how many cases were localized at the time of this information and subsequently showed metastasis.  Since the number of cases analyzed was too small, it is difficult to definitively conclude the results.
  • Previous report showed that TP53 negative prostate cancer cells were revealed possessing p53 mutation (PMID 14663467). The authors should also discuss about the issue.

Author Response

We wish to sincerely thank Reviewer #2 for their constructive comments and suggestions. We have addressed them as follows:

The title “predominantly Black patient cohort” is not scientific and confusing. The authors should focus on African American cohort only and revise the title.

The ambiguity that may be perceived with the previous title is understandable. We believe the best remedy is to avoid mention of any specific race or ethnicity in the title itself. For this reason, we have reworded the second half of the title to “…Findings from a Cohort of U.S. Military Veterans.” As alluded to in our resubmission letter, we believe it would be counterproductive to remove all non-African American patients from our present cohort. The reasons for this are two-fold:

  • Such an approach would unnecessarily decrease the size (n) of our cohort, reducing statistical power
  • Such a selective cohort would not represent a random selection of patients treated at our specific facility and undermine a key methodological strategy designed to lend credibility to our findings

Alternatively, one could argue to add new, additional African American patients, while removing non-African American patients, to retain, at a minimum, the current sample size. While this step would comply with such a suggestion and not jeopardize the validity of point 1), it would run counter to a fundamental principle of a study design relying on a random cohort selection. While we fully appreciate the incentive of a study design predicated on an exclusively African American cohort, at this junction, we believe that this study is fully justified in retaining a cohort that studies a representative cross-section of patients treated at our specific VA facility in Washington D.C. That our cohort is comprised of both African Americans and other ethnicities can be viewed as a strength in such a context. 

In terms of how best to denote the minority population in our study, we are more than willing to make the change from Black to African American. We wish to be culturally sensitive and want to ensure we use the most appropriate language.

The authors presented that they could not determine Grade Group of metastatic lesions. Why? Were the samples derived from post-hormonal treatment?

In the pathology literature and in our practice (VA, GW, MSKCC), metastatic deposits are not graded, nor graded by convention. However, we acknowledge the reviewer’s interesting idea in this era of personalized medicine and the possible new application of metastatectomies, for example. Metastatic deposits included in our investigation represented patients with de novo metastasis and thus did not receive prior therapy.

Table 1 should include information about samples derived from. How many cases were localized cases or metastatic cases?  The authors should also include information about the number of tissues derived from: RP, TUR, and biopsy.

We thank the reviewer for raising this point as it was not delineated in the initial manuscript. Included now in table 1 is information regarding disease stage (localized low risk - AJCC stage I, locally advanced - AJCC stage II/III, and metastatic disease - AJCC stage IV) and tissue specimen (radical prostatectomy, core biopsy, TUR/CHIPS, metastatic deposit). These findings are being discussed now in each corresponding paragraph. The presence of metastatic disease was found to be significantly associated with the presence of maximum (3+) p53 intensity. The presence of maximum (3+) p53 intensity was NOT associated with type of specimen. 

BCR depends on how long the patients are followed. Fisher’s exact test is not appropriate for the analysis.  The authors should use time-dependent analysis such as log rank test using KM survival curve.

BCR can be dependent on follow-up, and, in those situations, we agree that time-dependent analysis such as log-rank testing and Kaplan Meier survival curve analysis would be appropriate. However, in this dataset, BCR is a binary characteristic at baseline rather than a time-to-event outcome, making the log-rank testing inapplicable. Instead, Fisher’s exact test was used as an appropriate analysis for binary variables when the sample size is small. This point was discussed with our statistical team, and they confirmed that the use of Fisher’s exact test is appropriate in this setting.

The authors should present clinical and pathological information about cases included in Table 3.

We agree that this information was missing in table 3 of the original manuscript. In the revised version, we have now included in table 3 information regarding Grade Group and disease stage. 

“Our observation that maximum nuclear intensity was detected only in patients who developed metastatic disease, regardless of type of specimen (core biopsy, RP, TURP, or distant metastatic deposit), highlights the potential of this marker to inform about the severity and likelihood of progression of disease independent of other pathology-related factors” This information is important but we should know how many cases were localized at the time of this information and subsequently showed metastasis.  Since the number of cases analyzed was too small, it is difficult to definitively conclude the results.

We agree that this is important information, and it has now been incorporated into the discussion section. We have added information in table 1 regarding how many cases were localized low-risk, locally advanced and metastatic at time of diagnosis. In addition, we discuss how many patients with localized disease eventually developed metastatic disease during follow-up. While we agree the sample size is moderate, and precludes definitive conclusions, the study does provide strong evidence of an association, which by itself is valuable clinically for both patients and their health care providers.

Previous report showed that TP53 negative prostate cancer cells were revealed possessing p53 mutation (PMID 14663467). The authors should also discuss about the issue.

The existence of prostate cancer cells that are negative for p53 IHC despite the presence of an underlying TP53 mutation is an important observation. We thank the reviewer for bringing this to our attention and have now incorporated this information into the discussion section of the manuscript and the corresponding reference was added. We now raise the possibility that “silent” mutations exist, but also emphasize that these are rare and that their impact is unclear. We also acknowledge that this represents a potential limitation of p53 immunohistochemistry (IHC). While such limitations are to be expected when IHC is used as a screening tool, we feel this fact alone should not disqualify p53 IHC for this purpose.

This manuscript is a resubmission of an earlier submission. The following is a list of the peer review reports and author responses from that submission.

Round 1

Reviewer 1 Report

Comments and Suggestions for Authors

The study addresses an important aspect of prostate cancer prognosis by examining p53 nuclear expression, particularly in a predominantly Black cohort. This focus on an underserved population is commendable and fills a critical gap in current literature. The study's systematic evaluation of p53 intensity and percentage, supported by robust statistical analyses, provides a well-structured approach. The proposed scoring system for p53 overexpression is practical and has the potential for clinical application.

                  Limitations: The relatively small sample size (n=84) and the reliance on evaluations by only two pathologists limit the generalizability of the findings. Including a larger and more diverse cohort would strengthen the study's impact.

Molecular Validation while p53 IHC is presented as a surrogate for TP53 mutations, the limited sequencing data (available for only 29 patients) weakens this claim. Expanding the molecular validation across the entire cohort is strongly recommended. The emphasis on intensity as a prognostic marker is well-justified; however, combining intensity with percentage may provide a more comprehensive predictive model. Future studies should explore this synergistic potential.

The study briefly mentions automated pathology as a future direction but does not elaborate on its implementation. Automated image analysis could mitigate inter-observer variability and enhance the reproducibility of p53 scoring.

                  Need a deeper discussion of the biological mechanisms underlying p53 overexpression in prostate cancer progression, particularly any racial differences in TP53 alterations and their implications for personalized treatment. 

 The authors should emphasize its integration into routine pathology practice alongside other established biomarkers and the sequencing data provides valuable insights into the correlation between TP53 mutations and p53 IHC expression, but its limited scope and reliance on a tabular format could be enhanced by expanding the sample size and presenting genomic alterations through a graphical visualization, such as mutation heatmaps or oncoplots, to better illustrate patterns and clinical relevance.

Reviewer 2 Report

Comments and Suggestions for Authors

This reviewer agrees that positivity of TP53 IHC in prostate cancer specimens is important to predict biological aggressiveness. Weakness of the study included various stage of the disease (localized to metastatic prostate cancer) and also evaluated various types of tissues (RP specimens to biopsy ones). This reviewer cannot agree to evaluate TP53 IHC expression of biopsy samples and RP specimens in the same level.   

Please focus on localized prostate cancer using biopsy or RP specimens or on metastatic prostate cancer using biopsy samples and increasing evaluating cases to draw more definitive conclusion.